# Activin A forms a non-signaling complex with ACVR1 and type II Activin/BMP receptors via its finger 2 tip loop

Senem Aykul[1], Richard A Corpina[1], Erich J Goebel[2], Camille J Cunanan[1], Alexandra Dimitriou[1], Hyon Jong Kim[1], Qian Zhang[1], Ashique Rafique[1], Raymond Leidich[1], Xin Wang[1], Joyce McClain[1], Johanna Jimenez[1], Kalyan C Nannuru[1], Nyanza J Rothman[1], John B Lees-Shepard[1], Erik Martinez-Hackert[3], Andrew J Murphy[1], Thomas B Thompson[2], Aris N Economides[1]*, Vincent Idone[1]*

[1]Regeneron Pharmaceuticals, Tarrytown, United States; [2]University of Cincinnati, Cincinnati, United States; [3]Michigan State University, East Lansing, United States

**Abstract** Activin A functions in BMP signaling in two ways: it either engages ACVR1B to activate Smad2/3 signaling or binds ACVR1 to form a non-signaling complex (NSC). Although the former property has been studied extensively, the roles of the NSC remain unexplored. The genetic disorder fibrodysplasia ossificans progressiva (FOP) provides a unique window into ACVR1/Activin A signaling because in that disease Activin can either signal through FOP-mutant ACVR1 or form NSCs with wild-type ACVR1. To explore the role of the NSC, we generated 'agonist-only' Activin A muteins that activate ACVR1B but cannot form the NSC with ACVR1. Using one of these muteins, we demonstrate that failure to form the NSC in FOP results in more severe disease pathology. These results provide the first evidence for a biological role for the NSC in vivo and pave the way for further exploration of the NSC's physiological role in corresponding knock-in mice.

*For correspondence:
aris.economides@regeneron.com
(ANE);
vincent.idone@regeneron.com (VI)

Competing interest: See
page 16

Reviewing editor: Karen Lyons,
UCLA Orthopaedic Hospital,
University of California, Los
Angeles, Los Angeles, United
States

## Introduction

Activin A is a protein that belongs to the TGFß/BMP family of ligands (**Hinck et al., 2016**). These ligands initiate signaling by driving the formation of heterotetrameric complexes of their cognate type I and type II receptors, and where a dimeric ligand brings together two type I and two type II receptors (**Yadin et al., 2016**). There are seven type I and five type II receptors that mediate signaling for this family of ligands. The choice of type I receptor in the signaling complex is the main determinant of which of two signaling pathways is activated: Smad1/5/8, when the type I receptors are ACVRL1, ACVR1, BMPR1A, or BMPR1B; and Smad2/3, when the type I receptors are ACVR1B, TGFBR1, or ACVR1C. Activin A utilizes ACVR1B as its type I receptor in conjunction with the type II receptors ACVR2A, ACVR2B, and, to a lesser extent, BMPR2 to signal through Smad2/3 (**Aykul et al., 2017**). The biological functions of Activin A have been investigated widely using multiple approaches (**Makanji et al., 2014**; **Namwanje and Brown, 2016**) including reverse genetics experiments wherein the gene encoding for Activin A – *Inhba* – has been 'knocked out' (**Archambeault and Yao, 2010**; **Matzuk et al., 1995**; **Pangas et al., 2007**). The results of those experiments have been largely interpreted in the context of Activin A acting as an agonist of ACVR1B to induce Smad2/3 or other signaling pathways (**Archambeault and Yao, 2010**; **Makanji et al., 2014**; **Namwanje and Brown, 2016**; **Pangas et al., 2007**).

Interestingly, ACVR1 was originally cloned as the type I receptor for Activin A (**Attisano et al., 1993**; **Tsuchida et al., 1993**), but the inability of Activin A to activate ACVR1, followed by the discovery that BMP7 activates it, led to a 'relabeling' of ACVR1 as a BMP receptor (**Macías-Silva et al.,**

*1998*). However, recent studies pinpoint a previously unrecognized property of Activin A: that it can engage the type I receptor ACVR1, to form an ACVR1•Activin A•type II receptor non-signaling complex (NSC) (*Hatsell et al., 2015*; *Olsen et al., 2015*). This NSC is unique in that its stoichiometry is identical to that of corresponding signaling complexes formed between the same receptors and BMPs, but also in that it is converted into a signaling complex in the ACVR1-driven genetic disorder fibrodysplasia ossificans progressiva (FOP). Excluding the special situation of FOP (see below), the NSC functions to tie down Activin A and render it unavailable for signaling, but also to tie down the type II receptors and ACVR1 and render them unavailable for engagement with BMPs, hence resulting in an apparent inhibition of ACVR1-mediated BMP signaling; in cells where ACVR1 is the main type I receptor, Activin A inhibits BMP6- (*Hatsell et al., 2015*) and BMP7-induced signaling (this work).

The picture is more complex in FOP. FOP is a rare autosomal-dominant genetic disorder that arises from missense mutations in the sequence encoding the intracellular domain of ACVR1 (*Katagiri et al., 2018*). The most clinically important feature of FOP is the episodic, yet progressive and cumulative, formation of heterotopic bone in connective tissue, a process referred to as heterotopic ossification (HO) (*Hüning and Gillessen-Kaesbach, 2014*). FOP-mutant variants of ACVR1 display the neomorphic property of recognizing Activin A (as well as other Activins) as an agonistic ligand, much like a BMP (*Hatsell et al., 2015*; *Hino et al., 2015*). In mouse FOP, activation of FOP-mutant ACVR1 by Activin A is required for HO, as demonstrated by experiments where inhibition of Activin A, using highly specific monoclonal antibodies, halts both the occurrence and the progression of HO (*Hatsell et al., 2015*; *Lees-Shepard et al., 2018a*; *Lees-Shepard et al., 2018b*; *Upadhyay et al., 2017*). Thus, in FOP, the ACVR1[FOP mutant]•Activin A•type II receptor complex, which is stoichiometrically identical to the NSC, acts as a signaling complex. However, since FOP is autosomal-dominant, one wild-type copy of ACVR1 remains operational and hence capable of sequestering Activin A in NSCs. In mouse FOP, removal of the wild-type copy of *Acvr1* exacerbates the degree of HO (*Lees-Shepard et al., 2018b*). These data are consistent with a model where loss of wild-type ACVR1 results in loss of the NSC and concomitant increase in the number of active complexes between Activin A and FOP-mutant ACVR1. Whereas these results indicate that the NSC is operant in vivo, they do not provide direct evidence for the NSC, and also do not inform whether there are other biological roles for the NSC.

To date, there has been no systematic exploration of the biological roles of the NSC either within or outside of FOP. Outside of FOP, exploration of the NSC has been limited to myeloma cells in culture, wherein Activin A protects them from BMP9-induced apoptosis (*Olsen et al., 2015*); whether a similar phenomenon takes place in vivo is currently unknown. In order to begin to address such knowledge gaps, we engineered a set of 'agonist-only' Activin A muteins that retain their ability to activate ACVR1B but fail to form the NSC with ACVR1. Using these muteins, we show that the finger two tip loop (F2TL) region of Activin A is critical for the formation of the NSC, and we obtain conclusive in vivo evidence that the NSC is required for tempering the degree of HO in mouse FOP. Our successful engineering and in vivo application of agonist-only Activin A to explore the role of the NSC in FOP sets the stage for further exploration of the wider physiological roles of the NSC using the corresponding knock-in mice.

## Results

### The ACVR1•Activin A•type II receptor complex does not transduce signal

Before exploring the NSC in detail, we first revisited our initial finding that Activin A can form a complex with ACVR1 and Type II receptor but cannot stimulate signaling by wild-type ACVR1 (*Hatsell et al., 2015*). In that initial report, we focused on Smad1/5/8 phosphorylation or a reporter-based surrogate readout of that event. Therefore, in order to exclude the possibility that the ACVR1•Activin A•type II receptor may transduce signal by utilizing signaling proteins other than Smad1/5/8, we examined whether treatment of ACVR1-expressing cells with Activin A would result in phosphorylation of a set of proteins that represent major signaling pathways activated by receptor kinases.

To set the stage for these experiments, we first established conditions that bias the readout of signaling to that mediated by ACVR1 while minimizing potential contributions from the other two type I receptors that transduce signal in response to Activin A, ACVR1B and TGFBR1. To achieve this, we utilized cells that overexpress ACVR1 and also employed two inhibitors: an anti-ACVR1B blocking antibody (MAB222) and the TGFBR1 kinase-selective inhibitor SD208. We fist validated these two inhibitors, demonstrating that SD208 inhibits Smad2/3 signaling induced by either Activin A or TGFB1, while leaving BMP6-induced signaling (via ACVR1 and Smad1/5/8) unaffected (*Figure 1A*, upper panel). Similarly, MAB222 inhibits Smad2/3 signaling induced by Activin A while leaving signaling induced by either TGFB1 or BMP6 unaffected (*Figure 1A*, lower panel). In this manner, we established conditions that guard against contributions to signaling by ACVR1B and TGFBR1 in response to Activin A, yet do not interfere with signaling mediated by ACVR1.

In order to examine a wide array of signaling pathways, we utilized a Human Phosphokinase Array as it allows the simultaneous interrogation of the phosphorylation state of 37 different signal transducing proteins representing signaling pathways activated by receptor kinases, and includes known 'alternative' signaling pathways activated by the BMP/TGFβ family of ligands, yet not limited to the

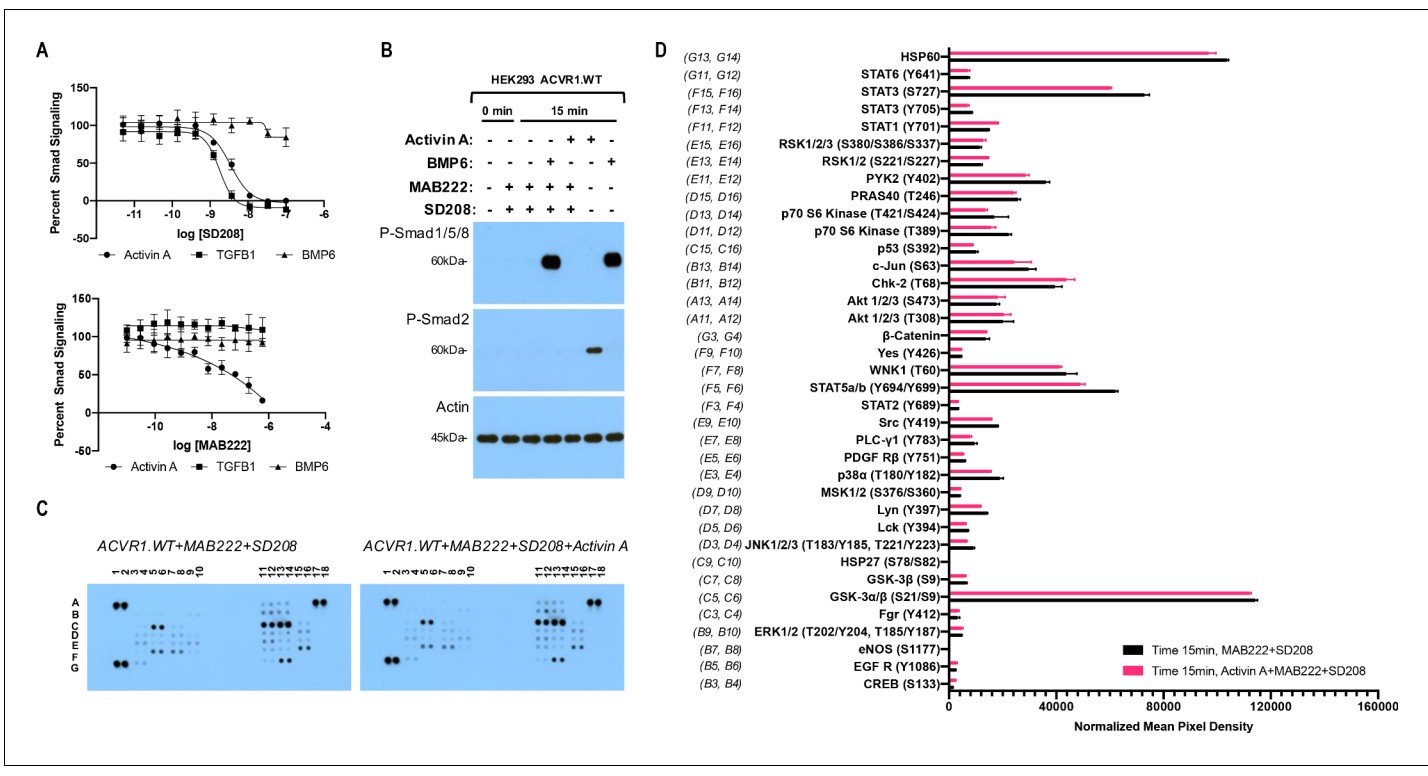

**Figure 1.** The Activin A•ACVR1•type II receptor complex does not transduce signal. (A) HEK293 cells harboring either a Smad2/3 or Smad1/5/8 reporter were treated with Activin A (1 nM), TGFβ1 (1 nM), or BMP6 (10 nM) in the presence of varying concentrations of SD208 or MAB222. (*top panel*) SD208 (TFGBR1 kinase inhibitor) inhibits Activin A-induced Smad2/3 signaling (IC$_{50}$: 3.2 nM) and TGFβ1-induced Smad2/3 signaling (IC$_{50}$: 1.4 nM) but does not affect BMP6-induced Smad1/5/8/signaling. (*bottom panel*) MAB222 (ACVR1B neutralizing antibody) inhibits Activin A induced Smad2/3 signaling (IC$_{50}$: 37.4 nM) but leaves TGFβ1-induced Smad2/3 and BMP6-induced Smad1/5/8 signaling unaffected. (B) Smad-mediated signaling of HEK293 cells overexpressing ACVR1 was analyzed by immunoblotting. MAB222 plus SD208 inhibit Activin A-induced Smad2/3 phosphorylation but not BMP6-induced Smad1/5/8 phosphorylation. Consistent with prior observations, Activin A does not induce Smad1/5/8 phosphorylation via wild-type ACVR1. (C) Membrane-based sandwich immunoassay analysis of kinase phosphorylation (RnD Systems Proteome Profiler Human Phosphokinase Array Kit) was applied to the same cellular lysates utilized on panel B. (D) Quantitative analysis of Human Phospho-Kinase Array blots shown in panel C. The Activin A•ACVR1•type II receptor complex does not directly activate downstream signaling of the pathways included in this panel, as evidenced by the lack of increases in any of the phosphoproteins assayed therein.

The online version of this article includes the following source data and figure supplement(s) for figure 1:

**Source data 1.** Reporter assay data of HEK293 cells treated with MAB222 or SD208, and Phospho-Kinase array data of HEK293 cells treated with MAB222 + SD208 in the presence and absence of Activin A.

**Figure supplement 1.** The Activin A•ACVR1•type II receptor complex does not transduce signal in mES cells.

latter. The phosphoproteins measured by this array are not preselected with transducers of BMP signaling, and hence represent an unbiased screen. In order to focus on the immediate signals transduced via ACVR1, we pretreated our HEK293 cells overexpressing wild type human ACVR1 with MAB22 plus SD208, and assessed phosphorylation of signal transducing proteins 15 min after addition of Activin A. This time frame is sufficient to induce phosphorylation of Smad2/3 by Activin A or Smad1/5/8 by BMP6, while 3 hr of pretreatment with MAB222 and SD208 is efficient at inhibiting signaling with ACVR1B and TGFBR1, leaving Smad1/5/8 signaling unperturbed (*Figure 1B*), and reducing the possibility of picking up secondary effects that usually arise at later time points. Comparison of the 37 proteins whose phosphorylation state can be interrogated using this array reveals of that there is no induction of phosphorylation by Activin A. The only significant differences detected were minor reductions in the phosphorylation level of STAT3 (S727) and STAT5 (*Figure 1C,D*). These results are consistent with the idea that the ACVR1•Activin A•type II receptor complex acts as an NSC: it does not transduce signal either through the Smads or any of the other proteins interrogated here – if anything, engagement of ACVR1 by Activin A suppresses signaling.

Effectively identical results were obtained when we repeated this assay using a different cell line, mouse ES cells, where ACVR1 is not overexpressed, and that are also homozygous-null for *Acvr1b*. (This alleviates the need for utilizing MAB222 to block interaction of Activin A with ACVR1B.) Consistent with the results shown in *Figure 1*, treatment of mES cells with Activin A in the presence of SD208 does not result in any increases in phosphorylation in the proteins interrogated by this array (*Figure 1—figure supplement 1*). This data further strengthens the notion that Activin A forms a NSC with ACVR1 and the corresponding type II receptors.

## A conceptual framework for engineering 'agonist-only' Activin A muteins

In order to set the stage for exploring the role of the ACVR1•Activin A•type II receptor NSC, we attempted to engineer Activin A muteins that:

1. Activate ACVR1B in a manner indistinguishable to wild type Activin A (*Figure 2A*, left panels).
2. Preserve Activin A's type I receptor preferences (i.e. avoid skewing towards increased utilization of other type I receptors such as TGFBR1 or ACVR1C).
3. Retain their interaction with Activin A's natural antagonists, Follistatin (FST) and Follistatin-like 3 (FSTL3) (*Figure 2A*, right panels).

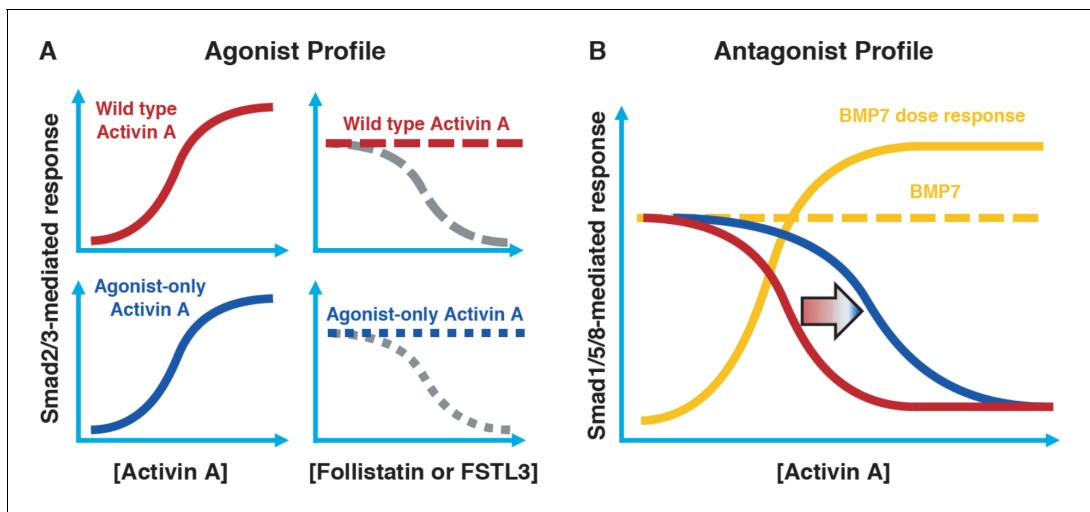

**Figure 2.** Conceptual framework for designing 'agonist-only' Activin A muteins. (**A**) Agonist only Activin A muteins (blue) should retain activation of the Smad2/3 pathway like wild type Activin A (red; left panels), and they should retain ability to be inhibited by the endogenous antagonists Follistatin and FSTL3 (grey; right panels). (**B**) As a result of a loss of ACVR1 binding, Activin A muteins should be less effective inhibitors of BMP-mediated signaling to the Smad1/5/8 pathway. Therefore, we expect that the agonist only muteins should have reduced antagonism of Smad1/5/8 signaling compared to wild-type Activin A. However, antagonism will not be entirely lost, as the agonist-only mutein must still bind to type II receptors for activation of Smad2/3 signaling via ACVR1B (and, to a lesser extent, TGFBR1).

4. Cannot form a NSC with ACVR1 as evidenced by a reduction of inhibition of BMP7-induced activation down to the level of inhibition that is driven solely by binding of Activin A to its type II receptors (*Figure 2B*). (Since binding of Activin A to type II receptors is a required property for signaling through ACVR1B [*Greenwald et al., 2004*], it follows that it is not possible to engineer an Activin A that signals through ACVR1B but does not compete with BMPs for type II receptor occupancy.)

As a first step to engineer Activin A muteins with such properties, we searched for BMP-TGFß family members that engage the same type II receptors as Activin A, utilize ACVR1B as their type I receptor, and do not engage ACVR1. Three ligands fulfill these requirements: Nodal, MSTN, and GDF11 (*Yadin et al., 2016*). Given that of these three, MSTN and GDF11 can also utilize TGFBR1 (*Andersson et al., 2006*; *Rebbapragada et al., 2003*), we focused on Nodal as the main donor of type I receptor-binding regions to substitute into the corresponding regions of Activin A, while also exploring the outcomes of substituting the same regions from other ligands. To define the specific regions of Activin to replace with their Nodal-derived counterparts, we initially relied on previously engineered artificial ligands where one region of Activin A had been replaced by the corresponding region from other BMPs (*Cash et al., 2009*; *Harrison et al., 2004*; *Korupolu et al., 2008*). Based on the premise that the regions that are most likely to define receptor utilization preferences are also those that are more evolutionarily divergent, we chose regions where the sequence diversity of TGFß ligands was significant.

More specifically, several studies have demonstrated that the pre-helix and post-helix regions make critical contacts with type I receptors. Substitution of the region encompassed by the pre-helix and post-helix region (amino acids 355 to 391 of mature Activin A) with the corresponding region from BMP2, switched type I (but not type II) receptor utilization (*Korupolu et al., 2008*), whereas swapping the pre-helix region of Activin A with that of MSTN switched the preference of Activin A from ACVR1B to TGFBR1 (*Cash et al., 2009*). Moreover, the recently solved structure of GDF11 (an Activin family ligand) in complex with TGFBR1 and ACVR2B (PDB:6MAC), shows that the finger two region is particularly important for determining type I receptor specificity for the Activin class (*Goebel et al., 2019*). Superimposition of Activin A into this ternary complex predicts that Activin A's finger two tip loop (F2TL; amino acids 406–409) also participates in binding to type I receptors (*Figure 3—figure supplement 1*). To further explore potential interactions of Activin A with ACVR1, we generated predictions of specific molecular interactions through structural modeling. We started with the structure of Activin A bound to Fs288 (PDB:2B0U) (*Thompson et al., 2005*), as Fs288 engages Activin A using the same interface as the type I receptor. This fixed the structure of Activin A as would be expected when bound to receptor. We then generated several models of ACVR1 based on closely related receptors and found BMPR1A to provide the best structural prediction. Superimposition of these models to the complex structure of TGFBR1•GDF11•ACVR2B followed by energy minimization of amino acid side chains suggested a key interaction at the fingertip. Specifically, D406 extends into the type I binding site (seen in several Activin A structures – PDB: 1S4Y, 2B0U, 2ARV, 3B4V) and is flanked by two lysine residues from ACVR1 (*Figure 3—figure supplement 1*; *Greenwald et al., 2004*; *Harrington et al., 2006*; *Stamler et al., 2008*; *Thompson et al., 2005*). This observation supports a key role for D406 in binding ACVR1. Our model is further supported by the fact that there is a conserved hydrophobic residue (M79 in ACVR1) that rests in a hydrophobic pocket formed by the Activin A dimer, forming a 'knob-in-hole' motif, which is also present in the BMP2:BMPR1A (*Allendorph et al., 2006*) and GDF11•TGFBR1 complexes (PDB:2GOO, 6MAC) (*Goebel et al., 2019*).

Along with the F2TL region, the pre- and post-helix regions also exhibit significant sequence diversity between ligands, hence making them attractive candidates for type I receptor selectivity (*Figure 3—figure supplement 2*). In contrast, other regions that are known to be involved in engaging the type I receptors, such as finger one and H3 helix, were excluded as donors for substitution analysis; finger one was excluded because it displays a high level of sequence conservation between ligands, whereas the H3 helix was not utilized because it appears to play a structural role and not to be a major determinant of type I receptor choice (*Allendorph et al., 2007*). Hence, we chose to focus on these three regions – F2TL, pre-helix, and post-helix – as sources of sequences to transfer from Nodal (and other Activin class ligands) into Activin A.

### Activin A with Nodal F2TL activates ACVR1B

Based on the design criteria and biochemical and structural information summarized above, we engineered Activin A muteins in which their pre-helix, post-helix, and F2TL regions were replaced with the corresponding regions of Nodal, generating Activin A.Nod.pre, Activin A.Nod.post, and Activin A.Nod.F2TL, respectively (*Figure 3*). After expression and purification of their mature forms (*Figure 3—figure supplement 3*), we first tested their ability to activate Smad2/3 signaling, a measure for utilization of ACVR1B as the type I receptor. Of these three muteins, the Activin A.Nod.F2TL exhibited activity very close to that of wild-type Activin A (*Figure 4*), whereas the other two were less active (*Figure 4—figure supplement 1*).

The realization that the tip loop variant performed according to our specifications for activation of ACVR1B prompted closer examination of the finger two tip loop region sequences between Activin A and Nodal. This comparison revealed an intriguing difference between the fingertips of these ligands: that Nodal lacks an aspartic acid at position 406 (*Figure 3B*). Moreover, an alignment of Activin A from different species reveals that this residue is highly conserved (*Figure 4—figure supplement 2*), whereas it varies when the different Activins are compared (*Figure 3—figure supplement 2*). Therefore, we tested whether a deletion of D406 would generate an Activin A mutein with properties similar to those of the Nodal tip loop variant (Activin A.Nod.F2TL). We compared the ability of Activin A.ΔD406, Activin A.Nod.F2TL, and wild type Activin A to activate ACVR1B using a CAGA-luciferase assay as a readout for Smad2/3 activation. As shown on *Figure 4A*, Activin A.Nod. F2TL is slightly less active than Activin A, whereas Activin A.ΔD406 exhibits identical activity to wild type Activin A, making Activin A.ΔD406 a second potential candidate for an agonist-only Activin A variant to study further.

### Activin A.Nod.F2TL and Activin A.ΔD406 retain their type I receptor preferences

In order to exclude the possibility that by substituting the sequence of finger two we have inadvertently altered type I receptor preferences (precluding the deliberate reduction of binding to ACVR1), we employed an assay that measures dimerization of type I with type II receptors and

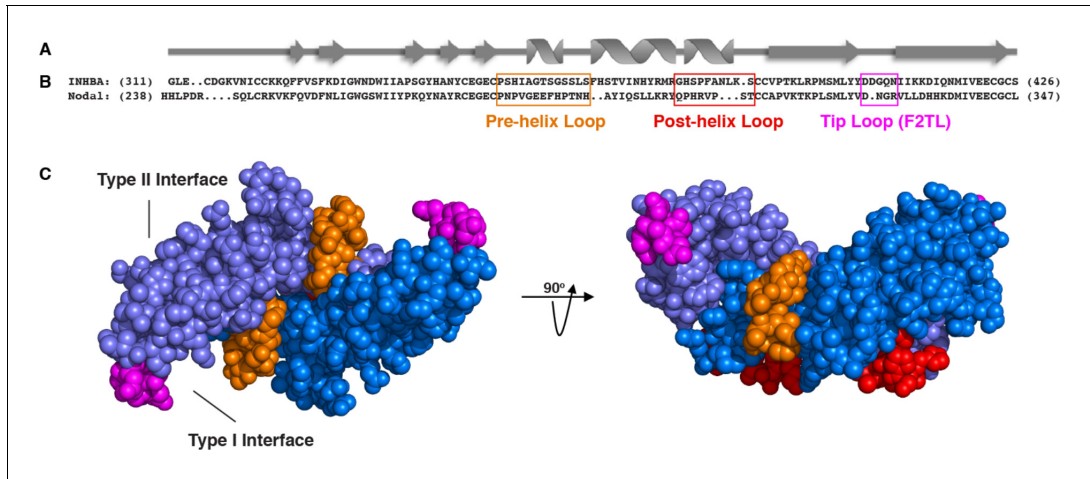

**Figure 3.** Activin A muteins were engineered wherein the Activin A pre-helix, post-helix, and F2TL regions were replaced with the corresponding regions of Nodal. (**A**) Ribbon diagram of Activin A. (**B**) Structural alignment of human Activin A and Nodal that highlights the Activin A pre-helix, post-helix and finger two tip loop (F2TL) sequences used to generate agonist-only Activin A muteins. Nodal sequences highlighted in the boxed areas were substituted for the corresponding sequences of Activin A. (**C**) Crystal structure of FSTL3-bound Activin A (space filled model, PDB 3B4V) with substituted areas colored as follows: pre-helix loop (orange), post-helix loop (red) and F2TL (magenta). Each Activin A monomer is depicted in either light or dark blue.

The online version of this article includes the following figure supplement(s) for figure 3:

**Figure supplement 1.** Model of Activin A:ACVR1 structure.

**Figure supplement 2.** Sequence alignment of mature ligands in human TGFβ family.

**Figure supplement 3.** Purified Activin and Activin A F2TL muteins.

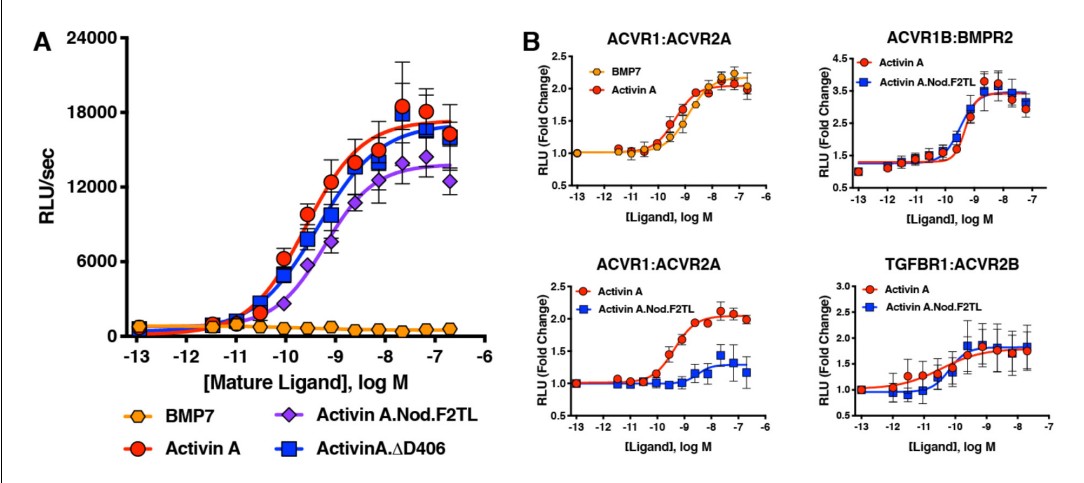

**Figure 4.** Activin A F2TL mutants signal normally through the Smad2/3 pathway, but display greatly reduced binding to ACVR1. (**A**) Activin A F2TL muteins activate Smad2/3 signaling to similar levels as wild-type Activin A in HEK293 cells harboring a Smad2/3 reporter using firefly luciferase. (**B**) U20S cells expressing split beta-galactosidase fusions of corresponding type I and type II receptors were treated with a dose response of BMP7, Activin A, or Activin A.Nod.F2TL. Type I receptor binding was measured by luminescence in these receptor dimerization assays. Activin A.Nod.F2TL has reduced ability to dimerize ACVR1:ACVR2A receptors, while retaining wild type capacity to dimerize the ACVR1B:BMPR2 and TGFBR1:ACVR2B receptor pairs. The data presented are representative of at least three independent biological replicates. Three technical replicates were performed per experiment. The online version of this article includes the following source data and figure supplement(s) for figure 4:

**Source data 1.** Reporter assay data of HEK293 cells and dimerization assay data of U20S cells treated with Activin A and Activin A F2TL muteins.
**Figure supplement 1.** Activin A with pre-helix and post-helix substitutions from Nodal display reduced ability to activate the Smad2/3 pathway.
**Figure supplement 2.** Sequence alignment of Activin A from various species.
**Figure supplement 3.** Activin A.GDF8.F2TL mutein binds TGFBR1 better than Activin A.

investigated whether Activin A.Nod.F2TL and ΔD406 muteins displayed altered type I receptor preferences. In line with our bioassay results, both Activin A.Nod.F2TL and Activin A.ΔD406 are identical to Activin A in their ability to dimerize ACVR1B and TGFBR1 along with the corresponding type II receptors (*Figure 4B*; *Figure 5—figure supplement 1*). In addition to these experiments, we also attempted to detect a complex of Activin A with ACVR1 using Surface Plasmon Resonance (SPR). Although we could detect binding of Activin A to ACVR1B or binding of a diverse set of BMPs to ACVR1, we were unable to detect binding of either Activin A alone or in complex with ACVR2A or ACVR2B or BMPR2 to ACVR1 (*Figure 5—figure supplement 2*). These results indicate that although Activin A can engage ACVR1 into a complex with type II receptors, that complex is not stable enough to be isolated in vitro. This is in line with prior observations that crosslinking is required to detect binding of Activin A to ACVR1 (*Attisano et al., 1993*; *Piek et al., 1999*; *Tsuchida et al., 1993*).

## Activin A.Nod.F2TL cannot form the NSC

In order to determine whether Activin A.Nod.F2TL and Activin A.ΔD406 have lost their ability to form the NSC with ACVR1 and type II receptors, we tested their capacity to inhibit BMP7-initiated activation of ACVR1 using a Smad1/5/8-responsive luciferase-based assay. Consistent with previous observations utilizing BMP6 (*Hatsell et al., 2015*) and BMP9 (*Hatsell et al., 2015*; *Olsen et al., 2015*), Activin A was able to inhibit activation of ACVR1 by BMP7 in dose-dependent manner. In contrast, both Activin A.Nod.F2TL and Activin A.ΔD406 were approximately 60-fold and 15-fold less effective at inhibiting BMP7, respectively (*Figure 5*; *Figure 5—figure supplement 3*). In addition, as would be expected for any Activin A variant that has reduced ability to form the NSC, both Activin A.Nod.F2TL (*Figure 4B*) and Activin A.ΔD406 (*Figure 5—figure supplement 1*) display greatly reduced ability to induce dimerization of ACVR1 with ACVR2A.

To exclude the possibility that this reduced ability to inhibit BMP7-initiated activation of ACVR1 is due to loss or reduction of binding of these muteins to type II receptors, we tested Activin A.Nod.F2TL and Activin A.ΔD406 for their ability to engage ACVR2B, one of the main type two receptors,

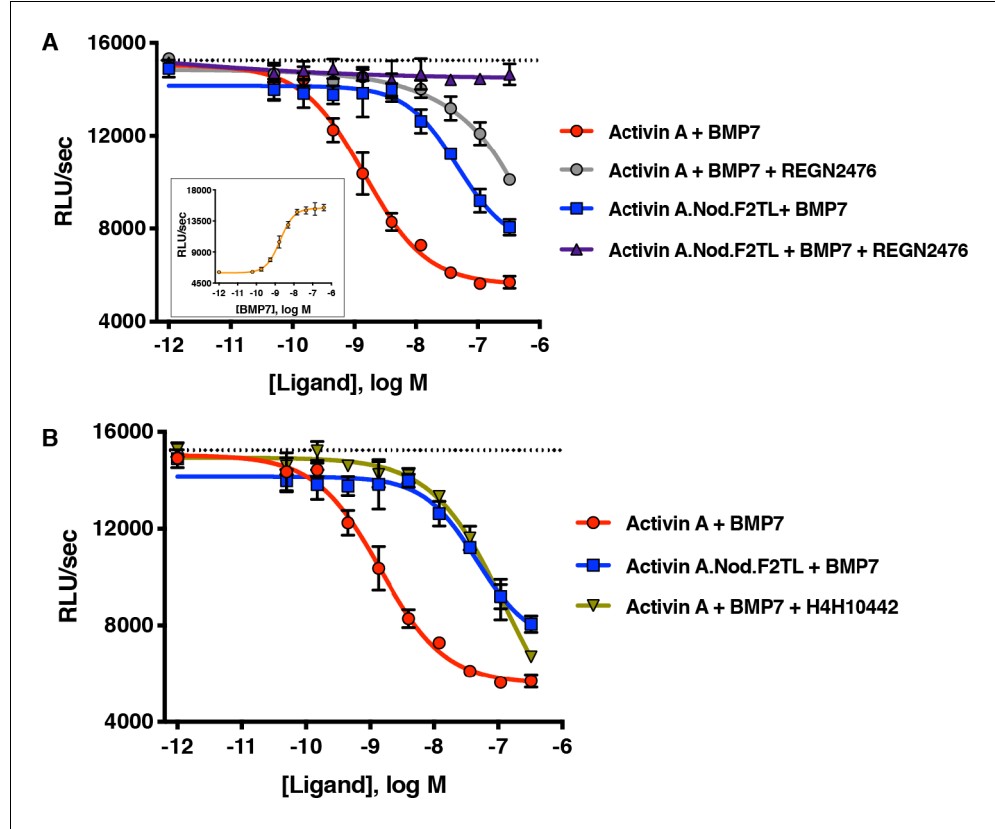

**Figure 5.** Activin A.Nod.F2TL fails to form a NSC with ACVR1 and corresponding type II receptors. (**A**) Activin A. Nod.F2TL is a less effective inhibitor of BMP7 signaling to Smad1/5/8 than wild-type Activin A. HEK293 cells harboring a Smad1/5/8 luciferase reporter construct were treated with varying concentrations of Activin A or Activin A.Nod.F2TL with a constant concentration of BMP7 (12 nM) to stimulate Smad1/5/8 signaling. Inhibition of BMP7 is reduced ~60 fold with Activin A.Nod.F2TL compared to Activin A. Using an Activin A antibody that blocks interaction with type II receptor (REGN2476), the remaining inhibition of BMP7 by Activin A.Nod.F2TL is lost. (**B**) Inhibition of type I receptor binding of Activin A with anti-Activin antibody H4H10442 shows a similar reduction in BMP inhibition to Activin A.Nod.F2TL. (The $IC_{50}$s of Activin A and Activin A.Nod.F2TL are $1.4 \times 10^{-9}$ M and $9.7 \times 10^{-8}$ M, respectively. Insert in panel A shows a dose response of BMP7 on the HEK293 reporter cells, and the dotted lines represents the Smad1/5/8 signal induced by 12 nM BMP7 without inhibition by Activin A.). The data presented are representative of at least three independent biological replicates. Three technical replicates were performed per experiment.

The online version of this article includes the following source data and figure supplement(s) for figure 5:

**Source data 1.** Reporter assay data of HEK293 cells treated with anti-Activin A antibodies in the presence of BMP7+Activin A or BMP7+Activin A.Nod.F2TL.

**Figure supplement 1.** Activin A F2TL muteins lose binding to ACVR1.

**Figure supplement 2.** Activin A•ACVR1 complex cannot be detected using SPR.

**Figure supplement 3.** Activin A.ΔD406 is 15-fold less effective than Activin A at inhibiting BMP7-induced Smad1/ 5/8 signaling through ACVR1.

**Figure supplement 4.** Activin A finger two tip loop muteins do not display altered type II receptor binding.

**Figure supplement 5.** Anti-Activin A antibody REGN2476 blocks binding of Activin A to type II receptors ACVR2A and ACVR2B.

**Figure supplement 6.** The anti-Activin A blocking antibody H4H10442 inhibits Activin A by binding to its finger two tip loop (F2TL) region.

**Figure supplement 7.** Activin B and Activin AB form a NSC with ACVR1.

using Biacore. In this binding assay, Activin A.Nod.F2TL and ActivinA.ΔD406 bound with affinity that is very similar to wild-type Activin A (*Figure 5—figure supplement 4*), confirming that the corresponding finger two tip loop substitutions in these two muteins do not affect type II receptor binding, and consistent with the fact that the F2TL region of this family of ligands is not involved in type

II receptor binding, but rather is a key determinant of type I receptor choice (*Goebel et al., 2019*). Furthermore, REGN2476, an anti-Activin A monoclonal antibody that neutralizes Activin A by blocking its binding to its type II receptors (*Figure 5—figure supplement 5*), completely alleviates inhibition of BMP7 signaling by Activin A.Nod.F2TL further indicating that the remaining inhibition is a result of binding of this mutein to type II receptors (*Figure 5A*). Similar results were obtained for ActivinA.ΔD406 in this assay (*Figure 5—figure supplement 3*); however, its reduced effectiveness in inhibiting BMP7-initiated activation of ACVR1 indicates that ActivinA.ΔD406 can still interact with ACVR1 to form a functional NSC, albeit not as well as wild-type Activin A.

To demonstrate that it is indeed the inability of Activin A.Nod.F2TL to form the NSC that accounts for its reduced ability to antagonize BMP7-initiated signaling by ACVR1, we used a neutralizing anti-Activin A antibody, H4H10442, that inhibits the interaction of Activin A with its type I receptors by binding to the finger two tip loop region (*Figure 5—figure supplement 6*). We hypothesized that binding of H4H10442 to Activin A will mimic the effect of the Nod.F2TL substitution, and hence result in a BMP7 signaling inhibition profile identical to that of Activin A.Nod.F2TL. Indeed, H4H10442 shifts the dose response of the inhibition brought about by wild-type Activin A on BMP7-initiated signaling to that obtained by Activin A.Nod.F2TL (*Figure 5B*), hence verifying that the reduced inhibition obtained with Activin A.Nod.F2TL is a direct consequence of the inability of Activin A. Nod.F2TL's F2TL region (which is that of Nodal) to bind ACVR1. Using the same criteria, our data shows that Activin A.ΔD406 has partially retained the ability to antagonize BMP7 signaling via ACVR1, making it a less desirable 'agonist-only' Activin A mutein (*Figure 5—figure supplement 3*). Taken together, these results clearly indicate that the remaining ability of Activin A.Nod.F2TL to antagonize BMP7-initiated signaling is indeed through engagement of the type II receptors, whereas its reduced ability to antagonize BMP7 is a property of its inability to engage ACVR1 to form the NSC.

Incidentally, in contrast to what is seen with Activin A.Nod.F2TL and Activin A.ΔD406, inhibition of binding of wild-type Activin A to the type II receptor (via REGN2476) does not 'flat-line' the ability of wild-type Activin A to inhibit BMP7-initiated signaling (*Figure 5*; *Figure 5—figure supplement 3*). In fact, the response to the two

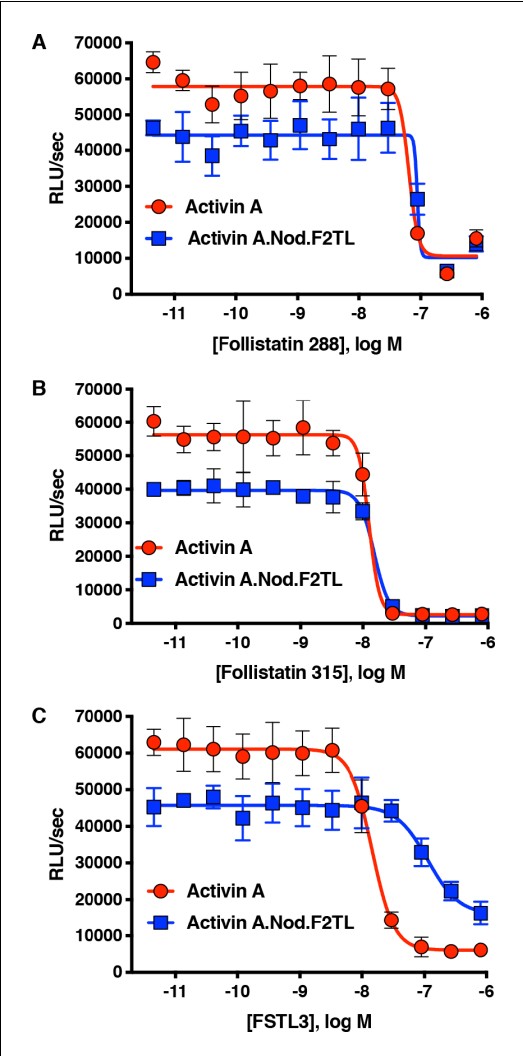

**Figure 6.** Activin A.Nod.F2TL mutein is inhibited by Follistatin but shows reduced inhibition by FSTL3. Varying concentrations of Follistatin and FSTL3 were preincubated with a constant concentration (10 nM) of Activin A or Activin A.Nod.F2TL 'agonist only' mutein. Activity of both Activin A and Activin A.Nod.F2TL were tested in HEK293 cells harboring the Smad2/3 luciferase reporter. Activity of both Activin A and Activin A.Nod.F2TL was blocked by both follistatin-288 (**A**) and follistatin-315 (**B**). (**C**) FSTL3 is a less effective inhibitor of Activin A.Nod.F2TL. The data presented are representative of at least three independent biological replicates. Three technical replicates were performed per experiment.

The online version of this article includes the following source data and figure supplement(s) for figure 6:

**Source data 1.** Reporter assay data of HEK293 cells treated with different isoforms of Follistatin in the presence of either Activin A or Activin A.Nod.F2TL.

**Figure supplement 1.** Activin A F2TL muteins are inhibited by Follistatin but show reduced inhibition by FSTL3.

antibodies - H4H10442, which blocks binding to ACVR1 by occupying the Activin A F2TL (*Figure 5—figure supplement 6*), and REGN2476, which blocks binding to the type II receptors (*Figure 5—figure supplement 5*)– is nearly identical when Activin A is the ligand outcompeting BMP7. This indicates that Activin A can engage ACVR1 even in the absence of binding to type II receptors, at least under conditions where ACVR1 is expressed at high levels.

## Activin A.Nod.F2TL and Activin A.ΔD406 retain their interactions with their antagonists

As a last step in examining potential differences between Activin A and these two muteins, we tested their interaction with natural Activin inhibitors, Follistatin (FST) and Follistatin-like 3 (FSTL3) (*Figure 6*). Both isoforms of FST, differing in the absence (Fs288, *Figure 6A*) or presence (Fs315, *Figure 6B*) of C-terminal acidic tail, inhibit Activin A signaling to baseline in Smad2/3 reporter assays. Activin A.Nod.F2TL displays nearly identical inhibition profile compared to wild type, whereas Activin A.ΔD406 shows a slightly decreased ability to be inhibited by both FST isoforms but still can be fully inhibited to baseline (*Figure 6—figure supplement 1*). Contrasting this, both F2TL muteins display decreased ability to be inhibited by FSTL3 (*Figure 6C*; *Figure 6—figure supplement 1*). We attribute this to a key difference in the way FSTL3 engages Activin A compared to FST. Although both FST and FSTL3 engage the Activins and other TGFß ligands in a 2:1 stoichiometric ratio with two Follistatin molecules per one dimeric ligand, when FST engages Activin A, the FSD3 domain reaches around the ligand to non-covalently contact the N-terminal domain of the other FST molecule, essentially fully wrapping around the ligand. (There are relatively few specific contacts between FST and the ligands that it interacts with; this makes FST a quite promiscuous modulator of multiple BMP/TGFß family members.) In contrast, FSTL3 lacks the FSD3 domain and engages Activin A using several specific contacts, which include direct contacts with the finger two tip loop. Specifically, the ring nitrogen of His91 in FSTL3 makes a charge contact with D406 of Activin A, and both Ser221 and Arg225 directly interact electrostatically with Activin A Q408 (*Cash et al., 2012*). Hence, the reduced inhibition of these Activin A muteins by FSTL3 is likely a consequence of their both lacking D406.

## Activin A FT2L muteins still activate ACVR1[R206H]

Given that both Activin A.Nod.F2TL and Activin A.ΔD406 display a greatly reduced ability to engage ACVR1, we tested whether they can still activate ACVR1[R206H] (i.e. the most common FOP-causing variant, which utilizes Activins as agonists [*Hatsell et al., 2015*; *Hino et al., 2015*]). Both muteins were found to transduce a signal through ACVR1[R206H] (*Figure 7—figure supplement 1*). This result is consistent with the observation that both muteins can still dimerize ACVR1 with ACVR2A to some extent, though not to the degree that Activin A does (*Figure 4B*; *Figure 5—figure supplement 1*), and also not to the degree required to form NSCs (*Figure 5*). In line with these results, structural modeling of Activin A.Nod.F2TL suggests that D405 can shift into the type I interface to compensate for the loss of D406, albeit interacting with only one lysine residue (K78), and hence resulting a weakened interaction (*Figure 3—figure supplement 2*). Furthermore, we found that in this assay Activin A.Nod.F2TL activates ACVR1[R206H] more than wild-type Activin A, a finding consistent with the inability of Activin A.Nod.F2TL to form the NSC.

## Activin A FT2L mutein results in increased HO in mouse FOP

Given that Activin A.Nod.F2TL escapes recruitment into NSCs yet can activate ACVR1[R206H] (presumably by transient engagement of ACVR1 – see *Figure 4B*), we hypothesized that in an FOP setting Activin A.Nod.F2TL will generate more heterotopic bone than wild type Activin A. To test this hypothesis, we induced HO by implantation of Activin A or Activin A.Nod.F2TL adsorbed collagen sponges into *Acvr1*[R206H]FlEx/+ (*Acvr1*[tm2.1Vlcg/+]; MGI:5763014) mice (FOP mice). Consistent with previous observations, FOP mice implanted with sponges containing Activin A develop HO by 2 weeks following implantation (*Hatsell et al., 2015*). As would be expected for any Activin mutein that escapes recruitment into the NSC (yet still activates ACVR1[R206H]), we observed significantly more HO in collagen sponges containing Activin A.Nod.F2TL compared to wild-type Activin A, in fact approximately six times greater (*Figure 7A,B*). These observations were confirmed using fibro/adipogenic progenitors (FAPs), which are the cells that form HO in FOP (*Dey et al., 2016*; *Lees-*

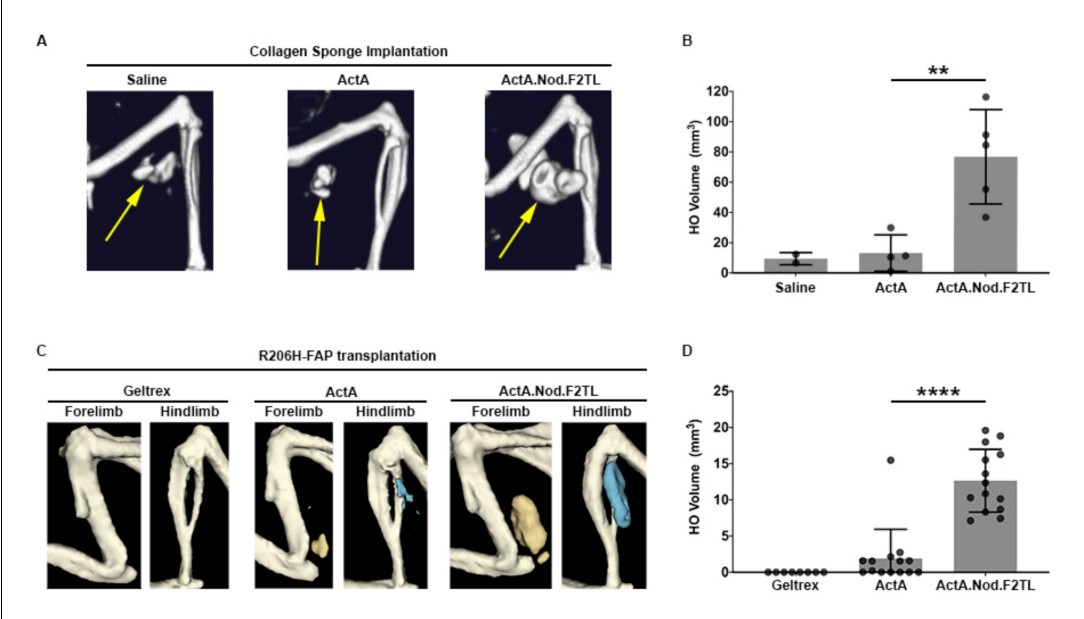

**Figure 7.** Activin A.Nod.F2TL mutein is a more potent inducer of HO than wild-type Activin A. (**A**) Representative μCT images of HO (yellow arrows) from tamoxifen-treated *Acvr1[R206H]FlEx/+; Gt(ROSA26)Sor^CreERT2/+* mice taken 2 weeks after implantation of collagen sponges adsorbed with saline, 20 μg wild-type Activin A (ActA), or 20 μg Activin A.Nod.F2TL (ActA.Nod.F2TL). (**B**) Quantification of HO volume 2 weeks post-implantation. Saline, n = 2; ActA, n = 4; ActA.Nod.F2TL, n = 5. Each dot represents a single implantation with group mean (grey bar) and ± standard deviation (error bars) shown. Statistical significance was assessed by one-way ANOVA; **=p ≤ 0.01. (**C**) Representative μCT images of fore- and hindlimbs from SCID hosts 11 days post-transplantation of FOP FAPs (R206H-FAPs) in Geltrex alone, Geltrex contain 5 μg ActA, or Geltrex containing 5 μg ActA.Nod.F2TL. HO is pseudocolored beige for forelimbs and blue for hindlimbs. (**D**) Quantification of HO volume 11 days post-transplantation of R206H-FAPs. Geltrex only, n = 8; ActA, n = 14; n = 14; ActA.Nod.F2TL. Each dot represents a single transplantation with group mean (grey bar) and ± standard deviation (error bars) shown. Statistical significance was assessed by one-way ANOVA; ****=p ≤ 0.0001.

The online version of this article includes the following figure supplement(s) for figure 7:

**Figure supplement 1.** Activin A F2TL muteins activate ACVR1[R206H] like wild-type Activin A.

*Shepard et al., 2018b*). FAPs from FOP mice were transplanted intramuscularly into SCID mice along with either Activin A or with Activin A.Nod.F2TL. When FOP FAPs were co-delivered with Activin A.Nod.F2TL, the resulting HO was also approximately six times the volume of HO obtained from FOP FAPs transplanted in the presence of wild-type Activin A (*Figure 7C, D*). These results are consistent with a model where Activin A.Nod.F2TL escapes recruitment into the NSC (with wild type Acvr1) and hence is free to engage the FOP-causing Acvr1[R206H] receptor more frequently than wild-type Activin A, simply because the latter can be trapped in the NSC, whereas Activin A.Nod.F2TL cannot. This increased frequency in engagement of Acvr1[R206H] by Activin A.Nod.F2TL results in increased signaling in the FAPs, and increased HO. These results are the first conclusive evidence that the NSC is operational in vivo where it negatively regulates the signaling that drives HO in FOP. They are also consistent with previous observations where deletion of wild-type *Acvr1* results in massively increased HO in FOP mice (*Lees-Shepard et al., 2018b*).

## Discussion

The discovery of the ACVR1•Activin A•type II receptor NSC was surprising and unprecedented. Unlike engagement of ligands or receptors by non-signaling or signal-suppressing decoy receptors (*Bonecchi et al., 2016*; *Boyce and Xing, 2007*; *Mishra et al., 2012*; *Onichtchouk et al., 1999*; *Palomo et al., 2015*; *Sheikh and Fornace, 2000*), or inhibitory ligands that naturally form 'dead-end' complexes with signal-transducing receptors (*Kokabu et al., 2012*; *Makanji et al., 2014*), the ACVR1•Activin A•type II receptor NSC is unique in that it stoichiometrically mimics signaling complexes utilizing the same type I and type II receptors, albeit with a different (not Activin) ligand class – the BMPs. Even more surprising was the finding that the FOP-causing missense mutations of

ACVR1 (which localize in the intracellular domain of this receptor) convert the NSC into a signaling complex, much like that formed by BMPs. The molecular mechanism that underlies this conversion is a matter of debate (*Hatsell et al., 2015*; *Hino et al., 2015*). That aside, the realization that the NSC effectively reduces the availability of all the partners involved in that complex (Activin A, ACVR1, and the type II receptors) and hence inhibits or reduces signaling by BMPs via the same receptors, propelled us to ask whether the NSC has biological roles. This question has, thus far, been largely unexplored as the majority of studies on Activin A have focused on its role as a signal-inducing ligand via ACVR1B (*Makanji et al., 2014*; *Namwanje and Brown, 2016*).

In order to explore the biological role(s) of the NSC in vivo, we engineered agonist-only variants of Activin A (i.e. variants that retain their ability to signal through ACVR1B but fail to form the NSC with ACVR1). To design such muteins, we made use of biochemical information regarding receptor utilization by Activin family ligands, prior mutagenesis studies, and structural modeling. We chose Nodal as the main donor for replacements of type I-binding regions of Activin A, a choice driven by the fact that Nodal utilizes the same type II receptors as Activin A along with ACVR1B but does not engage ACVR1. We then focused on three regions for engineering the different chimeric ligands: the pre-helix, the post-helix, and the finger two tip loop (F2TL) (see Results section). Of the resulting Activin A-Nodal chimeras, Activin A.Nod.F2TL – a mutein where the finger two tip loop was substituted with that of Nodal – displayed the desired properties: it activates ACVR1B nearly to the same degree as wild-type Activin A, does not switch to other type I receptors, and escapes engagement into the NSC. In addition, guided by structural models (*Figure 3—figure supplement 1*), we engineered a single amino acid deletion variant, Activin A.ΔD406. This variant also displayed reduced ability to form the NSC, indicating the importance of D406 as a determinant of the interaction of Activin A with ACVR1.

In parallel with our finding that the F2TL region is required for the formation of the NSC, the importance of the F2TL as a major determinant of type I receptor choice was concurrently demonstrated for GDF11's interaction with TGFBR1 (*Goebel et al., 2019*). Both findings are further supported by an additional chimeric ligand that we engineered using GDF8 as the 'donor' to generate the corresponding F2TL variant – Activin A.GDF8.F2TL. This variant utilizes TGFBR1 as the type I receptor to a greater degree than Activin A, while retaining its ability to engage ACVR1B (*Figure 4—figure supplement 3*). Therefore, our results lend further credence to the concept that the finger two tip loop is a key determinant of type I receptor choice in the Activin class.

Our findings show that the NSC is mediated through very specific contacts between Activin A and ACVR1. Nonetheless, we note that we did not attempt an exhaustive search to define all possible Activin A muteins with the desired properties, and hence it is possible that other such muteins can be engineered. Along these lines, since Activin B and Activin AB are also capable of forming NSCs with ACVR1 and the corresponding type II receptors (*Figure 5—figure supplement 7*), our findings should enable exploration of the formation of the respective complexes by following similar methodology.

Both of the two muteins that we study in detail here deviate from our initial design specifications in that they display a small increase in resistance to inhibition by one of Activin's natural antagonists, FSTL3 (*Figure 6*). This property is unavoidable for Activin A variants that bear changes in the finger two tip loop, as that region is involved in one of the main contacts between Activin A and FSTL3. Whether this deviation will have physiological consequences, remains to be seen. It should be noted, however, that FSTL3 knockout mice have a relatively benign phenotype that has been well characterized and that has been attributed to an increase in Activin A activity (*Mukherjee et al., 2007*). Given the relatively small reduction in the ability of Activin A.Nod.F2TL to be inhibited by FSTL3 (compared to what would be expected from a complete loss of FSTL3), it is unlikely that this difference will have phenotypic consequences. Nonetheless, the availability of phenotypic data on FSTL3 knockout mice should facilitate the assessment of any differences that might arise in the physiology of mice bearing Activin A.Nod.F2TL as a knock-in into *Inhba*.

The development of an agonist-only Activin A mutein, Activin A.Nod.F2TL, enabled us to directly query the role of the NSC in FOP, and hence build upon the prior discovery of the role of Activin A in this disease. Because of the ability of Activin A.Nod.F2TL to escape recruitment into NSCs, and hence not allow sequestration of both itself and the type II receptors into inactive complexes that render them inaccessible to participate in signaling, we predicted that Activin A.Nod.F2TL should precipitate more HO compared to wild type Activin A. We demonstrate this to be the case in two

different yet complimentary settings. These results provide the first evidence for a role of the NSC in vivo. Specifically for FOP, they also provide the first direct evidence that the NSC tempers HO in FOP, effectively reducing the amount of heterotopic bone that forms (presumably by reducing the availability of Activin A and type II receptors to form signaling complexes with FOP-mutant ACVR1). Our data is in agreement with the observation that deletion of wild type *Acvr1* in a mouse FOP setting results in massively increased heterotopic bone (*Lees-Shepard et al., 2018b*). The fact that Activin A.Nod.F2TL was successfully employed to query a pathophysiological process in vivo sets the stage for future experimentation with the corresponding knock-in mouse line outside of FOP. Given the level of evolutionary conservation in this receptor-ligand system, our data opens the field of investigating whether the ability of forming the NSC was an early or a late event during evolution.

## Materials and methods

### Antibodies and protein reagents
Activin A antibodies H4H10442 and REGN2476 are human monoclonal antibodies specific to Activin A and utilizing an IgG4 constant region. They were generated using Regeneron's VelocImmune mice (*Macdonald et al., 2014*; *Murphy et al., 2014*).

### Cell lines
HEK293 and U2OS cell lines were purchased from ATCC and their identity was confirmed by STR profiling. ACVR1[R206H]FlEx/+ and ACVR1[R206H]FlEx/+, ACVR1B KO mES cell lines were generated with VelociGene technology at Regeneron. Their identity was confirmed by NGS and TaqMan gene expression assay. All the cell lines used in this manuscript tested negative for mycoplasma.

### Cloning, expression and purification of Activin A muteins
The mature regions of Activin A or Activin A muteins were cloned downstream of mouse *Inhba* propeptide using isothermic Gibson assembly (*Gibson et al., 2010*) of synthetic gene blocks (Integrated DNA Technologies) into a CMV-based mammalian expression vector. The corresponding expression vectors were introduced into CHO-K1 cells expressing human Furin (CHO-KI.Furin). The resulting conditioned media was concentrated approximately ten-fold using a filtration column with a 3K molecular weight cut off (Pierce) before biochemical characterization of the Activin A muteins in cell-based assays. Following this primary analysis, Activin A and the Activin A F2TL mutants were stably expressed in CHO-K1.Furin cells. The Activin A pro-peptide complex was first purified to homogeneity by heparin chromatography followed by size exclusion chromatography on a preparative scale S200 column. To isolate the mature Activin A mutein dimers, reverse phase chromatography on C4 column was used, and mature Activin A was eluted with an acetonitrile gradient. The sequence information of Activin A muteins is available on Dryad (doi:10.5061/dryad.5hqbzkh3d).

### Structural modeling and alignments
ClustalW alignment in MacVector was used for amino acid sequence alignments. Structural alignments using known crystal structures as a reference were carried out using STRAP, a web-based alignment program (*Gille et al., 2014*). All figures with structural models were generated in PyMol (The PyMol Molecular Graphics System, Schrödinger, LLC, New York, NY). The model of ACVR1 was built using Swiss-model (*Waterhouse et al., 2018*). After manual adjustment that resulted in no clashes and contained realistic contacts, energy minimization of amino acid side chains was performed utilizing YASARA (*Krieger et al., 2009*).

### Binding kinetics measurements
Kinetic binding parameters for the interaction of human ACVR2B.Fc fusion protein with recombinant Activin A and Activin A muteins were determined under neutral pH on Biacore 3000 using dextran-coated (CM5) chips at 25°C. The running buffer was prepared using filtered HBS-EP (10 mM Hepes, 150 mM NaCl, 3.4 mM EDTA, 0.05% polysorbate 20, pH 7.4). The capture sensor surface was prepared by covalently immobilizing Protein A (Sigma-Aldrich, St. Louis, MO) to the chip surface using (1¬Ethyl-3-[3-dimethylaminopropyl]carbodiimide-hydrochloride)/N-hydroxysuccinimide (EDC/NHS) coupling chemistry. Binding kinetics of Activin A and Activin A muteins were measured by flowing

100nM-0.14 nM of ligand, serially diluted three-fold, at 100 uL/minute for two minutes and monitored for dissociation for 15 min. All capture surfaces were regenerated with one 30 s pulse of 10 mM glycine–HCl (pH 1.5, (GE Healthcare, Marlborough, MA). Kinetic parameters were obtained by globally fitting the data to a 1:1 binding model with mass transport limitation using Scrubber 2.0 c Evaluation Software. The equilibrium dissociation constant (KD) was calculated by dividing the dissociation rate constant (kd) by the association rate constant (ka).

## Cell culture and cell-based assays

Unless otherwise noted, cells were cultured in Dulbecco's modified Eagle's medium containing 10% (v/v) fetal bovine serum (FBS), penicillin/streptomycin (50 U/ml), and 2 mM L-glutamine. Generation of ACVR1 and ACVR1.R206H expressing HEK293 BRE-luciferase reporter cells has been previously described (*Hatsell et al., 2015*). For this study, a reporter construct with twelve tandem repeats of the $(CAGACagc)_{12}$ response element was introduced into HEK293 cells to engineer the HEK293 CAGA-luciferase reporter line (*Latres et al., 2015*).

Cell based assays were conducted as previously described (*Hatsell et al., 2015*). Briefly, HEK293 reporter cells were plated in 96-well plates. After a 16 hr incubation with ligands alone, ligand mixtures or ligand/antagonist mixtures, luciferase expression was measured with Bright-Glo Luciferase Assay System (Promega).

Human U2OS cell lines for receptor dimerization assays were purchased from DiscoverX (Fremont, CA) and used following manufacturers protocols. Briefly, cells were treated with ligands for 16 hr. β-galactosidase enzyme fragment complementation was quantified by a luminescent signal following the addition of the manufacturer-supplied substrate.

## Immunoassay analysis of kinase phosphorylation

In order to isolate signaling induced by Activin A only to the complex that it can form with ACVR1, HEK293 cells expressing ACVR1 were pretreated with 100 nM MAB222 (RnD Systems) plus 20 nM SD208 (BioVision) for 3 hr after overnight starvation. Then, signaling was induced either with 1 nM Activin A or 10 nM BMP6 for 15 min, in the presence of SD208 and MAB222. Cellular lysates were prepared by using lysis buffer (supplied in RnD Systems Proteome Profiler Human Phosphokinase Array Kit supplemented with 1X 'Protease Arrest' and 2X 'Phosphatase Arrest' (G-Biosciences)). Total protein concentration was determined by BCA kit (Thermoscientific). For Western blot, equal amounts of protein (20 μg) were separated under reducing conditions on 4–12% Novex WedgeWell gels (Invitrogen) and transferred to PVDF membrane (Advansta). Membranes were blocked with Superblock (Thermofisher) and incubated with primary antibodies from Cell Signaling at a 1:1000 dilution (anti-phospho-Smad2/3 (138D4), anti-phospho-Smad1/5/8 (41D10)), or 1:5000 (anti-β-actin (8H10D10)), followed by an incubation with horseradish peroxidase-conjugated secondary antibody at a 1:5000 dilution (7074). Western Bright ECL HRP substrate was used for detection (Advansta). Membrane-based sandwich immunoassay analysis of kinase phosphorylation (RnD Systems Proteome Profiler Human Phosphokinase Array Kit, ARY003B) was applied to the same cellular lysates utilized in phospho-Smad analysis. The kit was utilized according to manufacturer's protocol. Quantitative analysis of Human Phospho-Kinase Array blots was performed using Image J. Integrated densities of the spots were determined and normalized against control spots. Two independent biological and technical replicates were performed.

## In vivo implantation of wild type Activin A- or Activin A.Nod.FT2L mutant-containing collagen sponges

Collagen sponge containing recombinant proteins were implanted as described (*Hatsell et al., 2015*). Briefly, 4 mm diameter collagen sponges (SpongeCol, Advanced BioMatrix, Carlsbad, CA) were adsorbed with 10 μl suspension of either 20 μg of rhActivin A (R and D Systems, Minneapolis, MN), 20 μg of Activin A.Nod.FT2L mutant protein, or saline for 30 min at room temperature prior to implantation. Mice were anesthetized with Xylazine and Ketamine. Their right gluteus region was shaved and wiped with alcohol and betadine. Muscle tissue was exposed by making a small incision on the skin and fascia. Muscle fibers were separated using blunt forceps to create a pouch. Collagen sponges were implanted into the muscle pouch and separated muscle fibers were sutured with 5.0 absorbable suture (Visorb, CP Medical, Inc, Norcross, GA). Skin incisions were closed using instant

tissue adhesive (Surgi-Lock 2°C, Meridian Animal Health, Omaha, NE). Mice were given two doses of analgesic, Buprenex at a 0.1 mg/kg dose. Two days after implantation, $Acvr1^{[R206H]FlEx/+}$; $Gt(ROSA26)Sor^{CreERT2/+}$ mice were injected with tamoxifen as described (Hatsell et al., 2015) to initiate the model. Heterotopic bone formation was assessed at 2 weeks after implantation by in vivo μCT imaging. Mice were assigned to groups without randomization.

## Fibro/adipogenic progenitor (FAP) isolation and expansion

Details of skeletal muscle dissection have been previously described (Lees-Shepard et al., 2018a). Dissected muscle was dissociated from tamoxifen treated $Acvr1^{[R206H]FlEx/+}$; $Gt(ROSA26)Sor^{CreERT2/+}$ mice as described (Hatsell et al., 2015) for tamoxifen regime using the Skeletal Muscle Dissociation Kit (Miltenyi Biotec, Bergisch Gladbach, Germany) and gentleMACS Octo Dissociator with heaters (Miltenyi Biotec), in accordance with manufacturer instructions. Following centrifugation at 300 g for 10 min at 4°C, the supernatant was discarded and the pellet was resuspended in growth media (Dulbecco's Modified Eagle Medium (DMEM; Life Technologies, Carlsbad, CA) with 50 U/mL Penicillin and 50 μg/mL Streptomycin (Gibco, Billings, MT) and 20% fetal bovine serum (FBS; Lot# 192K18; Avantor, Randor, PA)). Cells were then plated onto tissue culture flasks (Corning, NY).

Magnetic-activated cell sorting (MACS) was performed on single cells incubated with anti-mouse Pdgfra APC (clone APA5; eBioscience, San Diego, CA) to label FAPs, followed by magnetic labeling with an Anti-APC MicroBeads kit (Miltenyi Biotec). Cells were then applied to magnetic MS columns (Miltenyi Biotec), in accordance with manufacturer instructions. MACS-isolated FAPs were seeded at a density of 2000 cells/cm$^2$ onto tissue culture flasks (Corning) in growth media and maintained at 37°C in a humidified atmosphere at 5% $CO_2$. Media was changed every other day. Prior to transplantation, FAPs were treated with 100 nM $(Z)-4$-hydroxytamoxifen (Sigma-Aldrich) for 24 hr to enhance inversion of the R206H-containing exon.

## Transplantation and drug treatment

MACS-isolated and expanded FAPs were resuspended at $1 \times 10^7$ cells/mL in ice-cold 1X Dulbecco's Phosphate-Buffered Saline (DPBS; Gibco) or Geltrex (LDEV-Free hESC qualified Reduced Growth Factor; Gibco) with or without 2.5 μg Activin A or Activin A.Nod.F2TL, as indicated in the corresponding figure. A 50 μL volume containing $2 \times 10^5$ FAPs was injected into the gastrocnemius or triceps muscle of SCID mice (JAX Stock No: 001913, The Jackson Laboratory, Bar Harbor, ME), as previously described. Mice were assigned to groups without randomization.

## μCT imaging analyses and measurements

Heterotopic bone formation was assessed as described previously (Hatsell et al., 2015), Briefly, mice were anesthetized with isoflurane and whole body–scanned, with a field of view at 60 mm ×120 mm, using in vivo μCT (Quantum GX, PerkinElmer). The x-ray source was set to a current of 88 μA, voltage of 90 kVp, with a voxel size at 240 mm. The CT imaging was visualized and quantified using Analyze software (Analyze Direct, Overland Park, KS). Whole-body bone tissue was auto segmented by threshold, the heterotopic bone was manually segmented from the original bone, and then bone volume calculated using Analyze. HO volume following transplantation was quantified as previously described (Lees-Shepard et al., 2018b). Mouse group information was blinded for HO analysis.

## Statistical analysis

Statistical analysis was performed using GraphPad Prism (GraphPad, La Jolla, CA). All numerical values are presented as mean ± standard deviation. One-way ANOVA was used to determine significance, as described in the corresponding figure legends. Differences were considered significant at $p < 0.05$.

## Additional information

### Competing interests

Senem Aykul, Richard A Corpina, Camille J Cunanan, Alexandra Dimitriou, Hyon Jong Kim, Qian Zhang, Ashique Rafique, Raymond Leidich, Xin Wang, Joyce McClain, Johanna Jimenez, Kalyan C Nannuru, Nyanza J Rothman, John B Lees-Shepard, Andrew J Murphy, Aris N Economides, Vincent Idone: The author is an employee of Regeneron Pharmaceuticals, Inc. Regeneron is currently developing a monoclonal antibody that neutralizes Activin A (REGN2477) as a potential therapy in fibrodysplasia ossificans progressiva (see https://clinicaltrials.gov/ct2/show/NCT03188666). The other authors declare that no competing interests exist.

### Funding

The authors declare that there was no funding for this work.

### Author contributions

Senem Aykul, Data curation, Formal analysis, Validation, Investigation, Methodology, Writing - review and editing; Richard A Corpina, Methodology; Erich J Goebel, Investigation, Methodology, Writing - original draft, Writing - review and editing; Camille J Cunanan, Data curation, Formal analysis; Alexandra Dimitriou, Investigation, Methodology; Hyon Jong Kim, Data curation, Formal analysis, Investigation, Methodology; Qian Zhang, Johanna Jimenez, Nyanza J Rothman, Data curation, Investigation, Methodology; Ashique Rafique, Formal analysis, Methodology; Raymond Leidich, Xin Wang, Data curation, Formal analysis, Methodology; Joyce McClain, Data curation, Methodology; Kalyan C Nannuru, John B Lees-Shepard, Data curation, Formal analysis, Supervision, Investigation, Methodology, Writing - review and editing; Erik Martinez-Hackert, Resources, Investigation; Andrew J Murphy, Writing - review and editing; Thomas B Thompson, Data curation, Formal analysis, Supervision, Methodology, Writing - review and editing; Aris N Economides, Conceptualization, Data curation, Supervision, Investigation, Visualization, Methodology, Writing - original draft, Project administration, Writing - review and editing; Vincent Idone, Conceptualization, Resources, Data curation, Formal analysis, Supervision, Validation, Investigation, Visualization, Methodology, Writing - original draft, Project administration, Writing - review and editing

### Author ORCIDs

Senem Aykul https://orcid.org/0000-0001-7250-2913
Erich J Goebel https://orcid.org/0000-0001-5549-9425
Camille J Cunanan https://orcid.org/0000-0003-0022-6117
Hyon Jong Kim https://orcid.org/0000-0002-3106-4977
Ashique Rafique https://orcid.org/0000-0003-4078-7242
Xin Wang https://orcid.org/0000-0002-7687-6928
Johanna Jimenez https://orcid.org/0000-0001-5947-0558
Kalyan C Nannuru https://orcid.org/0000-0003-1386-3764
Nyanza J Rothman https://orcid.org/0000-0002-6167-4370
John B Lees-Shepard https://orcid.org/0000-0002-1275-5799
Erik Martinez-Hackert https://orcid.org/0000-0002-6182-7023
Andrew J Murphy https://orcid.org/0000-0003-4152-4081
Thomas B Thompson https://orcid.org/0000-0002-7041-5047
Aris N Economides https://orcid.org/0000-0002-6508-8942
Vincent Idone https://orcid.org/0000-0002-5622-2011

### Ethics

Animal experimentation: This study was performed in strict accordance with the recommendations in the Guide for the Care and Use of Laboratory Animals of the National Institutes of Health. All of the animals were handled according to approved institutional animal care and use committee (IACUC) protocols.

Decision letter and Author response
Decision letter https://doi.org/10.7554/eLife.54582.sa1
Author response https://doi.org/10.7554/eLife.54582.sa2

## Additional files

### Supplementary files

• Transparent reporting form

### Data availability

We are providing source files for the data shown in the main figures of the manuscript.

The following dataset was generated:

| Author(s) | Year | Dataset title | Dataset URL | Database and Identifier |
|---|---|---|---|---|
| Aykul S | 2020 | Data from: Activin A forms a non-signaling complex with ACVR1 and type II Activin/BMP receptors via its finger 2 tip loop | https://doi.org/10.5061/dryad.5hqbzkh3d | Dryad Digital Repository, 10.5061/dryad.5hqbzkh3d |

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
