## [Decision Letter]

**Acceptance summary:**

This paper is important because it demonstrates the importance of Activin A interactions with the type I BMP receptor ACVR1. Most studies have focused on the role of Activin A as an agonist for the type I receptor ACVR1B. The work in this paper uses elegant biochemical approaches and in vivo validation to demonstrate that Activin A binding to ACVR1 leads to a non-signaling complex that antagonizes signaling by BMP7, a well-characterized ligand for ACVR1. The discovery that some ligands exert important functions via formation of non-signaling complexes may extend to other members of the TGFβ/BMP superfamily.

**Decision letter after peer review:**

Thank you for submitting your article "The finger 2 tip loop of Activin A is required for the formation of its non-signaling complex with ACVR1" for consideration by *eLife*. Your article has been reviewed by three peer reviewers, including Karen Lyons as the Reviewing Editor and Reviewer #1, and the evaluation has been overseen by Kathryn Cheah as the Senior Editor. The following individual involved in review of your submission has agreed to reveal their identity: Takenobu Katagiri (Reviewer #2).

The reviewers have discussed the reviews with each other and the Reviewing Editor has drafted this decision to help you prepare a revised submission.

Summary:

This manuscript addresses interactions between Activin and the type I BMP receptor ACVR1 through the generation of Activin A muteins. The rationale for the study arises from the fact that ACVR1 was initially cloned as an Activin-binding receptor, but Activins do not activate ACVR1. Subsequently, it was found that ACVR1 binds specific BMPs and activates BMP pathways rather than Activin/TGFβ pathways. Interest in the role of the Activin/ACVR1 interaction was re-ignited upon the discovery by the authors and confirmed by others that Activin A acts as an antagonist of BMP binding to WT ACVR1, but has agonist activity through ACVR1R[206H], the mutation underlying most cases of FOP. The underlying mechanisms for this change in responsiveness to Activin A are still not understood. This paper generates mutant activins (muteins) of Activin A, specifically, "agonist only" muteins, as an approach to tease apart the role of Activin A as an agonist for Activin/TGFβ signaling via its canonical receptor ACVR1B from its role as an antagonist of BMP signaling through ACVR1. Two such muteins are described, and one, based on replacement of the F2TL region of Activin A with the corresponding region of Nodal, retained binding to and activation of ACVR1B, but had greatly impaired binding to ACVR1, hence losing its ability to antagonize signaling by other BMPs. The authors present a range of biochemical and in vitro assays to characterize the function of these muteins. The authors also test the function of one of the muteins in vivo as an agonist for ACVR1[R206H], and find that it leads to enhanced heterotopic ossification compared to WT Activin A. From these results, they make inferences about the existence and impact of an Activin A/ACVR1/type II receptor non-signaling complex (NSC). There is general consensus that the authors have achieved the goal of generating muteins that show greatly impaired binding to ACVR1 while retaining binding to and signaling through ACVR1B. The in vivo data are intriguing and will set the stage for more detailed mechanistic studies. In summary, this work highlights the importance of interactions of activin with ACVR1 for BMP signaling.

Essential revisions:

There are some points for which all reviewers agreed additional data are needed:

1) The authors should present direct biochemical data demonstrating reduced binding of ACVRI ectodomain to the muteins (compared to WT Activin A).

2) The split β-gal assays should be done comparing the binding of Activin A and the muteins on ACVR1 and ACVR1B in combination with one of the ACVR2 receptors (most logically, ACVR2A since that was used throughout the figures).

3) More information is needed about the generation and properties of the Regeneron antibodies in order for reviewers to assess the solidity of the conclusions. Along these lines, it is also important to show some of the actual surface plasmon resonance data and accompanying fits. Similarly, statistical tests and methodology are needed for the supplementary figures.

There were also some points to address in the text:

4) The authors need to address why Activin A did not induce heterotopic ossification (HO) in the FOP model (Figure 6) given their previously published data showing that Activin A induces HO.

5) With respect to the Discussion, please address the points raised by reviewer 3 regarding the presence of D406 in other BMP ligands, and relate this to their ability to engage ACVR1.

6) There was a discussion but not consensus over the definition of "non-signaling complex" and the extent to which this manuscript can make conclusions about this complex. There was good agreement that the paper focuses on the properties of the Activin A muteins. Therefore, a change in title to remove "non-signaling complex" is appropriate. Instead, consider something like "The F2TL is required for interaction of Activin A with ACVR1”.

7) Along the same lines as the previous comment, please take out references to the NSC in the Abstract. It is appropriate to discuss how the data relate to the existence and function of the NSC elsewhere in the text, but the above changes to the title and Abstract will more closely reflect the bulk of the work in the paper.

8) It is unclear how the mutein enhances heterotopic ossification in vivo, when Figure 5—figure supplement 1 states that the muteins lose binding to ACVR1. Figure 7—figure supplement 1 indicates that the muteins can signal through ACVR1[R206H]. A more thorough discussion of the interpretation of the relationship between Figure 5—figure supplement 1 and Figure 7—figure supplement 1, along with a discussion of other possible interpretations is warranted.

---

## [Author Response]

Essential revisions:There are some points for which all reviewers agreed additional data are needed:1) The authors should present direct biochemical data demonstrating reduced binding of ACVRI ectodomain to the muteins (compared to WT Activin A).

We have attempted to demonstrate binding of Activin A to ACVR1 using Surface Plasmon Resonance. However, after several attempts we have failed to demonstrate binding to ACVR1 of even wild type Activin A (or Activin B) either alone or in complex with the ectodomain of ACVR2A, ACVR2B, or BMPR2. In contrast, we were able to show binding of BMP6, 7, and 9 to ACVR1, as well as binding of Activin A to ALK4 (indicating that the reagents that we are using are active). This data is in agreement with unpublished attempts by other groups to visualize an Activin A•ACVR1 complex. The inability to measure binding of even wild type Activin A to ACVR1 consequently precludes interrogation of any differences in affinity that any of the Activin muteins that we have generated may have. We have added this negative (but well controlled) evidence in Figure 5—figure supplement 2 and have commented accordingly in the main text. Nonetheless, the data that we have already procured with the DiscovRx lines (split β-gal assays) clearly demonstrate that Activin A is very effective in heterodimerizing ACVR1 with ACVR2A, whereas Activin A.Nod.F2TL has largely lost that ability.

2) The split β-gal assays should be done comparing the binding of Activin A and the muteins on ACVR1 and ACVR1B in combination with one of the ACVR2 receptors (most logically, ACVR2A since that was used throughout the figures).

We agree with the reviewers that this would have been the preferred format. However, the split β-gal assay that combines ACVR1B with ACVR2A has not been generated by the company that engineers these assays. Therefore, in as much as we would like to, we cannot generate that data.

3) More information is needed about the generation and properties of the Regeneron antibodies in order for reviewers to assess the solidity of the conclusions. Along these lines, it is also important to show some of the actual surface plasmon resonance data and accompanying fits. Similarly, statistical tests and methodology are needed for the supplementary figures.

As we have described in the Experimental Procedures section, the anti-Activin A antibodies used in our experiments were generated using Regeneron’s VelocImmune mouse technology. That technology has been described in the corresponding references, also supplied in the original text (Macdonald et al., 2014; Murphy et al., 2014). The plasmon resonance data for the two antibodies has now been added to the Figure 5—figure supplement 5D. We have also added a description of the statistical tests and any additional methods for the supplementary material.

There were also some points to address in the text:4) The authors need to address why Activin A did not induce heterotopic ossification (HO) in the FOP model (Figure 6) given their previously published data showing that Activin A induces HO.

We understand this query to be about the apparent lack of a great difference between the collagen sponge implantations and specifically the comparison between Saline and Activin A. This is due to the fact that occasionally just the injury associated with the performing these implantations can cause HO at the site of implantation. Although this occurrence is rare, its incidence is not zero and in this particular experiment the two FOP mice implanted with collagen sponge without Activin A did form some heterotopic bone. We did not observe this in the cohort where Geltrex was used as the carrier, but we opted to include all of the data that we have procured from this line of experimentation. The data demonstrates that there is a very clear difference between Activin A and Activin A.Nod.F2TL, with the latter being much more effective at inducing HO than its wild type counterpart.

5) With respect to the Discussion, please address the points raised by reviewer 3 regarding the presence of D406 in other BMP ligands, and relate this to their ability to engage ACVR1.

We are unclear as to what we are being asked to address here, as no specific comments have been provided in this review with respect to D406’s presence in other ligands. Nonetheless, we note that D405 seems to be more highly conserved across multiple BMPs, whereas the identity of amino acid at the position corresponding to D406 in Activin A is more variable. However, we have not explored this further as the focus of our work has not been to explore the role of D406 across different BMPs; we have only focused on generating Activin A muteins that have “agonist-only” properties.

6) There was a discussion but not consensus over the definition of "non-signaling complex" and the extent to which this manuscript can make conclusions about this complex. There was good agreement that the paper focuses on the properties of the Activin A muteins. Therefore, a change in title to remove "non-signaling complex" is appropriate. Instead, consider something like "The F2TL is required for interaction of Activin A with ACVR1”.

We respectfully disagree with the reviewers assessment. Our manuscript is very much about the Non-Signaling Complex (NSC) formed between Activin A and ACVR1 (along with the corresponding type II receptors) and this is what we are setting out to explore. We have shown here as well as in our previously published work (Hatsell et al., 2015) that when Activin A engages ACVR1 in the context of the corresponding type II receptors, the resulting complex does not induce phosphorylation of Smad1/5/8, and that it also antagonizes BMP6-induced signaling. However, in order to provide additional evidence that the ACVR1•Activin A•type IIR complex does not transduce any other signal, we have utilized a phosphoarray panel that captures the activation of the main non-canonical pathways that have been observed for BMPs in the literature as well as a wider array of signaling proteins not yet implicated as direct transducers of BMP-induced signaling. We have included this additional data at the beginning of the Results section. As shown on what is now a new Figure 1 and Figure 1—figure supplement 1, there are no significant changes in the phosphorylation state of the signal transducing proteins that we can interrogated using a Human Phosphokinase Array – if anything, the only differences are consistent with inhibition of signaling. Therefore, we propose that these results, combined with other data shown here as well as the evidence present in our prior manuscript (Hatsell et al., 2015), demonstrate that the ACVR1•Activin A•type IIR complex does not transduce signal, and hence is appropriately labeled as a NSC. Given the centrality of this idea to our paper, we would like to retain the current Title and Abstract, and not dilute the main thesis of our work.

7) Along the same lines as the previous comment, please take out references to the NSC in the Abstract. It is appropriate to discuss how the data relate to the existence and function of the NSC elsewhere in the text, but the above changes to the title and Abstract will more closely reflect the bulk of the work in the paper.

We respectfully submit that we have already addressed this point in our reply to query #6, above.

8) It is unclear how the mutein enhances heterotopic ossification in vivo, when Figure 5—figure supplement 1 states that the muteins lose binding to ACVR1. Figure 7—figure supplement 1 indicates that the muteins can signal through ACVR1[R206H]. A more thorough discussion of the interpretation of the relationship between Figure 5—figure supplement 1 and Figure 7—figure supplement 1, along with a discussion of other possible interpretations is warranted.

We agree that this point is far from obvious. However, it should be clear that we state in the text that neither one of the two muteins we have generated has completely lost the ability to engage ACVR1: “The finger two tip loop muteins have reduced ability to dimerize ACVR1 with ACVR2A…” as stated in the figure legend and “both Activin A.Nod.F2TL (Figure 3B) and Activin A.∆D406 (Figure 5—figure supplement 1) display greatly reduced ability to induce dimerization of ACVR1 with ACVR2A”, as stated in the main text. (We have now revised the title of Figure 5—figure supplement 1 to make this point clear.) In light of the fact that we do not completely abrogate formation of an ACVR1•Activin A•Type II receptor complex, it is not surprising, that both of Activin A.∆406 and Activin A.Nod.F2TL can activate signaling of ACVR1[R206H], as shown on Figure 7—figure supplement 1. We think that the results we show here reinforce each other, and we are unable to come up with alternative explanations that can be supported by the data that we have generated and presented here.